# DIFFUSION MODELS ARE TRAINING-FREE OBJECT TRACKERS

## ABSTRACT

Diffusion models, though developed for image generation, implicitly capture rich semantic structures. We observe that their self-attention maps can be reinterpreted as semantic label propagation kernels, providing robust pixel-level correspondences between relevant image regions. Extending this mechanism across frames yields a temporal propagation kernel that enables zero-shot object tracking via segmentation without training. We further enhance this process with test-time optimizations: DDIM inversion for semantically aligned representations, textual inversion for object-specific cues, and adaptive head weighting to combine complementary attention patterns. To this end, we propose DRIFT, which combines cross-frame self-attention with test-time optimizations and achieves state-of-the-art zero-shot performance on standard VOS benchmarks, competitive with supervised approaches and underscoring the semantic capture ability of diffusion self-attention.

## 1 INTRODUCTION

Diffusion models (Sohl-Dickstein et al., 2015; Ho et al., 2020; Song et al., 2020) are a class of generative models that synthesize data by reversing a gradual noising process. Initially proposed for image generation (Rombach et al., 2022; Ramesh et al., 2022; Saharia et al., 2022; Podell et al., 2023), recent studies have shown that they also internalize surprisingly strong semantic structures (Tang et al., 2023; Hedlin et al., 2023; Tian et al., 2024; Couairon et al., 2024; Wang et al., 2024b; Zhu et al., 2024; Zhang et al., 2024). This phenomenon arises from the denoising objective itself: in order to recover a coherent image from noisy latents, the model must implicitly learn how different regions of the image relate to one another in a meaningful way.

Some prior studies demonstrate that intermediate diffusion features encode rich semantic information (Hedlin et al., 2023) or that cross-attention effectively localizes visual concepts by aligning visual and textual representations (Zhang et al., 2024). In contrast, our work centers on the self-attention layers, where pairwise similarities between query and key features are computed. As observed in (Wang et al., 2024a), self-attention maps can refine coarsely localized object regions into precise masks through multiplicative interactions, effectively serving as a label propagation module. Building on this observation, we extend the use of self-attention maps across multiple frames through cross-frame attention.

In this work, we show cross-frame attention transforms spatial attention into a temporal label propagation kernel, providing a strong foundation for object tracking via segmentation-without any task-specific training. To further enhance this process, we introduce three complementary test-time optimizations—DDIM inversion, mask-specific textual inversion, and adaptive head weighting—which make the propagated masks more accurate and object-aware. To this end, we propose DRIFT, a diffusion-based object tracking framework that combines cross-frame attention with three test-time optimizations. Extensive experiments on four standard VOS benchmarks demonstrate that DRIFT achieves state-of-the-art performance in zero-shot settings, with results that are even competitive with several supervised approaches.

Our contributions can be summarized as follows:

- By extending self-attention in diffusion models to multiple frames, we show that the resulting affinity maps function as a label-propagation kernel, enabling zero-shot object tracking.

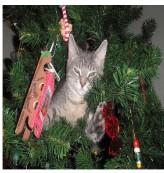 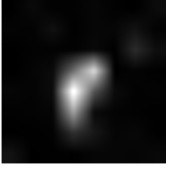 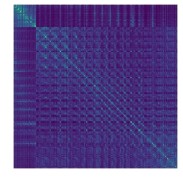 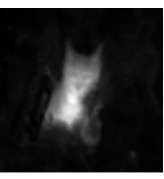

(a) Original image      (b) Coarse map      (c) Self-attention map      (d) Propagated mask

Figure 1: **Visualization of Label Propagation Using Features from a Text-to-Image Diffusion Models.** Given an input image (a), the coarse map, which corresponds to the cross-attention response for the token "cat" (b) provides approximate object localization based on the text prompt, while the self-attention map (c) captures semantic affinities within the diffusion model to refine this localization. Leveraging the self-attention map as a learned label propagation kernel, the final propagated mask (d) achieves substantially improved spatial precision.

- We show effectiveness of cross-frame attention with three test-time optimization on temporal label propagation with diffusion models.

- We conduct extensive experiments on four standard VOS benchmarks, where our framework consistently achieves state-of-the-art performance in the zero-shot setting.

## 2 RELATED WORK

**Zero-Shot Segmentation** Zero-shot segmentation aims to segment objects without task-specific training or class-specific supervision. In images, it is often formulated as open-vocabulary segmentation (Ding et al., 2022; Ghiasi et al., 2022; Liang et al., 2023; Zhou et al., 2022), where models align visual and language features (Radford et al., 2021) to enable class generalization, but typically require segmentation annotations during training. In contrast, recent diffusion-based methods (Couairon et al., 2024; Tian et al., 2024; Wang et al., 2024a) exploit the internal representaion of pretrained text-to-image diffusion models to perform segmentation without any segmentation-specific supervision. In videos, STC (Jabri et al., 2020) and DINO (Caron et al., 2021) use self-supervised visual features to propagate labels across frames based on spatial and temporal consistency. Building on these observations, we investigate whether the self-attention maps of pretrained diffusion models can also support temporal label propagation for zero-shot object segmentation via segmentation.

**Video Segmentation with Diffusion Models** Recent studies have leveraged pretrained diffusion models for object tracking by exploiting their strong generative priors. Diff-Tracker (Zhang et al., 2024) employs diffusion models in an unsupervised manner by using cross-attention to localize object regions, while a motion encoder and a target-specific prompt enable adaptation to object motion. VD-IT (Zhu et al., 2024) employs a text-to-video diffusion model for referring video object segmentation, which can also be viewed as a form of object tracking, and trains a segmentation head in a fully supervised manner. SMITE (Alimohammadi et al., 2025) addresses video part segmentation by fine-tuning the cross-attention layers of a diffusion model and further incorporates CoTracker (Karaev et al., 2024) to track segment points across frames. In contrast, our work demonstrates that the self-attention of pretrained diffusion models can be directly leveraged as a label propagation kernel, enabling standalone video object tracking without any video-based training or auxiliary modules.

## 3 METHOD

In this work, our primary goal is to show that pretrained text-to-image diffusion models can be repurposed as zero-shot, training-free object trackers. The key insight is that the pairwise query–key interactions in diffusion self-attention naturally support label propagation across frames, which forms the foundation of object tracking by segmentation. We further identify the self-attention mechanism as the central component enabling this propagation.

### 3.1 TEMPORAL LABEL PROPAGATION VIA CROSS-FRAME ATTENTION

Recent studies (Couairon et al., 2024; Tian et al., 2024; Wang et al., 2024a) demonstrate that pre-trained diffusion models can perform semantic segmentation without task-specific training. This ability arises from the cross-attention layers, which align text and visual tokens and thereby highlight visual regions corresponding to text-specified classes. However, the spatial maps produced by cross-attention alone are typically coarse. Accuracy improves only when these maps are multiplied by self-attention maps, as shown in Figure 1. This indicates that self-attention serves as a label propagation kernel: activations at one pixel can propagate to other pixels with similar semantics, refining a coarse mask into a more detailed segmentation. In this sense, the self-attention map functions as a learned mechanism for semantic label propagation. Formally, the self-attention map at layer $l$ is defined as $A_{\text{self}}^{(l,h)} = \text{softmax}\left(\frac{Q^{(l,h)} \cdot K^{(l,h)^\top}}{\sqrt{d}}\right)$, where $Q^{(l,h)}, K^{(l,h)} \in \mathbb{R}^{N \times d}$ are the query and key matrices of head $h$ in layer $l$, $N = H \times W$ is the number of spatial locations, and $d$ is the dimension per head. Averaging over all layers and heads yields $A_{\text{self}}$, which encodes semantic affinities between pixel pairs and serves as a propagation kernel that spreads coarse activations into fine-grained segmentation masks.

Building on this interpretation, we extend label propagation from the spatial domain of a single image to the temporal domain of a video. Given two consecutive frames $I_{t-1}$ and $I_t$, we compute a cross-frame attention map that measures similarities between features across frames:

$$\bar{A}_{t,t-1} = \sum_{l \in \mathcal{L}} \left( \sum_{h=1}^{H} w^{(l,h)} \text{softmax}\left( \frac{Q_t^{(l,h)} \cdot K_{t-1}^{(l,h)^\top}}{\sqrt{d}} \right) \right), \tag{1}$$

where $Q_t^{(l,h)}$ and $K_{t-1}^{(l,h)}$ are the query and key matrices from head $h$ of layer $l$ in frames $t$ and $t-1$, respectively, and $w^{(l,h)}$ is the weight assigned to each head (by default $w^{(l,h)} = \frac{1}{|\mathcal{L}| \times H}$). Each row of $\bar{A}_{t,t-1}$ defines how the label at a pixel in frame $t$ should aggregate information from frame $t-1$. The propagated mask is then updated as $\hat{M}_t = \bar{A}_{t,t-1} \hat{M}_{t-1}$.

In this way, the diffusion model's self-attention is repurposed as a cross-frame propagation kernel, enabling masks specified in the first frame to be consistently propagated through the video in a zero-shot manner.

### 3.2 RAW DIFFUSION FEATURE SIMILARITIES VS. SELF-ATTENTION MAPS

Many prior approaches (Jabri et al., 2020; Caron et al., 2021) often rely on cosine similarity of learned features for label propagation, and several recent studies have explored using pretrained diffusion features for this purpose (Couairon et al., 2024; Tang et al., 2023). In these works, the raw diffusion features are extracted and directly used as visual representations. However, we find that relying on raw diffusion features and measuring pairwise cosine similarity overlooks reusable pretrained knowledge embedded in the self-attention layers of diffusion models.

A self-attention map inherently captures feature similarity, but unlike cosine similarity, it does so after learned query and key projections that act as filters, preserving certain salient aspects for similarity. Moreover, multi-head self-attention incorporates multiple such projections, enabling the model to capture diverse semantic relationships and produce more robust similarity maps. In contrast, raw diffusion features—being optimized for image generation—may encode aspects that are irrelevant to semantic similarity. As a result, cosine similarity over these features often produces noisy maps dispersed across unrelated regions. The example in Figure 2 illustrates this supporting our arguments. Given two input frames (a), the cosine similarity of features (b) propagates the point in frame $t$ beyond the relevant region in frame $t'$, dispersing it across the entire image and even into irrelevant areas. As shown in (d), the multiple heads of self-attention each highlight different but still relevant regions, providing complementary views of semantic similarity. By aggregating these heads (c), the model effectively emphasizes the relevant region while suppressing spurious propagation.

### 3.3 TEST-TIME DIFFUSION OPTIMIZATION FOR LABEL PROPAGATION

Although diffusion self-attention inherently captures semantic correspondences that enable label propagation, its raw form is often insufficient for reliable mask propagation across frames. We there-

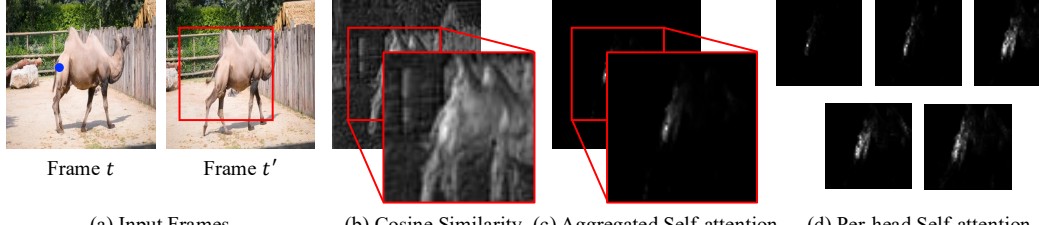

| Frame $t$ | Frame $t'$ | | |
|---|---|---|---|
| (a) Input Frames | (b) Cosine Similarity | (c) Aggregated Self-attention | (d) Per-head Self-attention |

Figure 2: **Comparison of Cosine Similarity vs. Self-attention for Label Propagation.** (a) A point in the frame $t$ is propagated to frame $t'$. (b) Cosine similarity produces dispersed activations scattered across unrelated regions. (c) The aggregated self-attention map, in contrast, focuses sharply on the target object region. (d) Individual attention heads exhibit complementary but distinct patterns, highlighting the diverse semantic relationships captured by multi-head self-attention.

fore investigate how to strengthen its propagation capability through three complementary test-time optimization techniques for diffusion models—DDIM inversion, mask-specific textual inversion, and adaptive head weighting—which together yield more accurate and object-aware masks.

### 3.3.1 DDIM INVERSION FOR SEMANTIC LATENT CONSTRUCTION

Diffusion models are inherently designed to take noisy images as input, but excessive or insufficient noise can distort semantics. At large timesteps, latents are dominated by noise and lose semantic information, while at very small timesteps, nearly noise-free latents provide weak semantic cues since the model has little incentive to predict noise accurately. Thus, constructing semantically meaningful representations requires injecting an appropriate amount of noise. We employ DDIM inversion (Song et al., 2020; Dhariwal & Nichol, 2021) as an effective solution. Instead of adding white noise, it perturbs images with model-predicted noise, aligning latents with the model's semantic manifold. This produces model-specific noisy inputs that preserve semantic structure, leading to more reliable cross-frame attention maps. While prior work has applied this technique to video editing (Cong et al., 2023), primarily to ensure faithful reconstruction of unedited content, our study demonstrates a distinct advantage for label propagation. Specifically, we show that DDIM inversion enables content-specific feature representations from a fully frozen diffusion model, producing attention maps particularly well-suited for propagating labels across frames.

### 3.3.2 MASK-SPECIFIC PROMPTS VIA TEXTUAL INVERSION

To exploit cross-frame attention maps from a text-to-image diffusion model for label propagation, the model requires a text prompt as input—something not naturally available in video inputs for object tracking. A naïve solution is to use a `null` prompt, but such prompts fail to capture information about the target object and its visual context. To address this, we adopt textual inversion (Gal et al., 2022), tuning a set of learnable text tokens specifically for mask propagation. Concretely, we compute an aggregated attention map $\bar{A}_{0,0}(\theta)$ from Equation (1) by feeding the initial frame at $t = 0$ as both query and key, which yields a self-attention map over the initial frame. The learnable text token embeddings $\theta$ are provided to the model alongside the input frame when computing this map. We then propagate labels on the initial frame using its GT mask $M_0$, $\hat{M}_0 = \bar{A}_{0,0}(\theta) \cdot M_0$.

Since propagation occurs within the same image, $\hat{M}_0$ should ideally reconstruct the original mask $M_0$, *i.e.*, the propagation kernel $\bar{A}_{0,0}(\theta)$ should spread labels consistently across the object region. To achieve this, we optimize $\theta$ with the following loss:

$$\mathcal{J} = \frac{1}{N} \sum_{i=1}^{N} \text{BCE}\left(\hat{M}_0(i), M_0(i)\right) \tag{2}$$

where BCE denotes binary cross-entropy and $M_0(i)$, $\hat{M}_0(i)$ are the values at the $i$-th spatial location in the GT and predicted masks, respectively.

Unlike prior work (Zhang et al., 2024), which applies textual inversion to cross-attention maps in order to improve the alignment of text embeddings with visual semantics, our approach directly

optimizes embeddings for label propagation in self-attention maps—making it closely aligned with object tracking. Interestingly, we find that the resulting embeddings are not semantically meaningful in the sense of corresponding to natural words; rather, they are scattered far from the modes of word embeddings in the text space. We further analyze these learned embeddings in Section 5.2.

### 3.3.3 ADAPTIVE WEIGHTING OF MULTI-HEAD ATTENTION

As discussed in Section 3.2, diffusion models employ multi-head self-attention, with each head capturing different semantic correspondences. Since some heads are more informative than others, we can replace uniform averaging with optimized head-specific weights $w^{(l,h)} \in [0,1]$, constrained such that $\sum_{l \in \mathcal{L}} \sum_{h=1}^{H} w^{(l,h)} = 1$. To this end, we can perform test-time optimization, updating these weights jointly with the mask-specific text embeddings $\theta$ by minimizing the loss in Equation (2). This allows informative heads to receive higher weights while less useful ones are down-weighted. With this refinement, the final attention map becomes a weighted aggregation of all heads, as defined in Equation (1), which can improve segmentation quality by emphasizing heads that capture stronger semantic correspondences.

## 4 DIFFUSION-BASED OBJECT TRACKING WITH MASK REFINEMENT

Building on the zero-shot label propagation capability of pretrained diffusion models, we introduce DRIFT (Diffusion-based Region Inference with cross-Frame Attention for Tracking), which achieves state-of-the-art performance in zero-shot object tracking via segmentation. Our approach combines a pretrained text-to-image diffusion model with the Segment Anything Model (SAM) (Kirillov et al., 2023) for mask refinement. The method operates operating in a fully training-free manner keeping all networks frozen. An overview of the full pipeline is shown in Figure 7.

### 4.1 ZERO-SHOT OBJECT TRACKING VIA SEGMENTATION

The proposed method tackles object tracking via segmentation in a zero-shot setting. Given a video, the input is an accurate mask of the target object in the first frame, and the goal is to generate precise segmentation masks that trace the object throughout the subsequent frames. In prior work (Pont-Tuset et al., 2017), this task is often described as semi-supervised video object segmentation, since the initial mask serves as a form of supervision. In contrast, we adopt the term object tracking via segmentation to avoid confusion with our zero-shot setup, where no task-specific training data are used. Formally, a video of $T + 1$ frames is denoted as $\mathcal{V} = \{I_0, I_1, \ldots, I_T\}$ with $I_t \in \mathbb{R}^{H \times W \times 3}$, and the provided first-frame mask is $M_0 \in \{0, 1, \ldots, X\}^{H \times W}$ for $X$ objects and background (label 0). The goal is to predict masks $M_t$ for $t = 1, \ldots, T$. Our method, DRIFT, addresses this task in a fully zero-shot, training-free manner by leveraging the inherent label propagation capability of pretrained text-to-image diffusion models discussed in the previous section.

### 4.2 MULTI-FRAME LABEL PROPAGATION

To improve robustness in label propagation, DRIFT extends cross-frame attention to multiple preceding frames (Caron et al., 2021; Jabri et al., 2020). For each target frame $t$, we compute attention maps $\{\bar{A}_{t,t-s}\}_{s=1}^{S}$ and restrict them spatially with a radius-$r$ mask. We further sparsify by keeping the top-$k$ scores across frames and normalizing them, encouraging stable correspondences. The final mask is obtained by aggregating propagated masks as $\hat{M}_t = \sum_{s=1}^{S} \bar{A}_{t,t-s} \hat{M}_{t-s}$. In the multi-object setting (Pont-Tuset et al., 2017), object and background masks are propagated independently, and the final segmentation is obtained by a pixel-wise argmax, $M_t = \arg\max_{x \in \{0,\ldots,X\}} \hat{M}_t^{(x)}$.

### 4.3 MASK REFINEMENT WITH SAM

While the cross-frame attention mechanism already enables zero-shot object tracking without any task-specific training, the predicted mask quality can be further enhanced by integrating SAM. To do this, we treat each soft mask prediction $\hat{M}_t^{(x)}$ as a spatial probability distribution by normalizing it so that the pixel values sum to one. This normalized distribution serves as a strong prior for the target object's location and shape. Based on this, we sample $p$ sets of $n$ point prompts from the

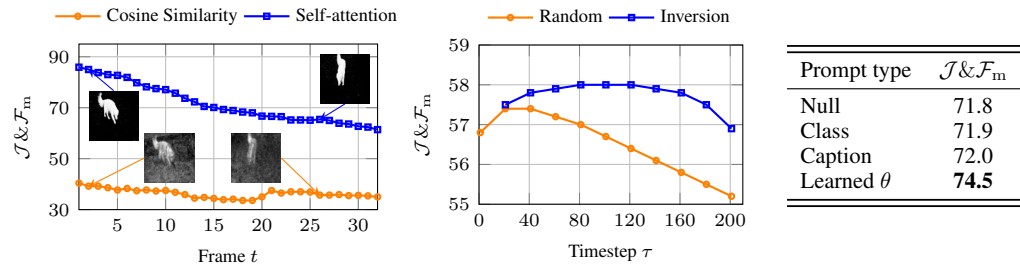

(a) Self-attn. vs. cosine sim. over $t$      (b) Random vs. Inversion across $\tau$      (c) Effect of prompt type.

| Prompt type | $\mathcal{J}\&\mathcal{F}_{\mathrm{m}}$ |
|---|---|
| Null | 71.8 |
| Class | 71.9 |
| Caption | 72.0 |
| Learned $\theta$ | **74.5** |

Figure 3: **Analyses of Label Propagation and Text Prompt Types.** (a) Per-frame $\mathcal{J}\&\mathcal{F}_{\mathrm{m}}$ scores for the first 30 frames, comparing self-attention–based and cosine-similarity–based affinity maps with mask visualizations. (b) $\mathcal{J}\&\mathcal{F}_{\mathrm{m}}$ scores across diffusion timesteps under random noise injection and DDIM inversion. (c) DRIFT results with different prompt types: null prompt, object class names, BLIP-2–generated captions, and learned embeddings. All results are reported on DAVIS 2017.

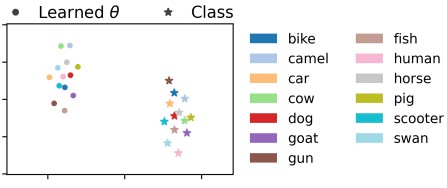

(a) t-SNE of learned $\theta$ and class words.

| Method | $w^{(l,h)}$ | $\mathcal{J}\&\mathcal{F}_{\mathrm{m}}$ |
|---|---|---|
| Baseline | Uniform | 71.1 |
| | Learned | 71.3 |
| +DI+TI | Uniform | 74.5 |
| | Learned | 74.8 |

(b) Effect of attention head weighting.

Figure 4: **Analyses of Learned Textual Embeddings and Attention Head Weighting.** (a) t-SNE visualization of embeddings, comparing textual-inversion–learned embeddings and object class word embeddings. (b) $\mathcal{J}\&\mathcal{F}_{\mathrm{m}}$ under uniform and learned attention head weighting, evaluated both without and with DDIM inversion and textual inversion. All results are reported on DAVIS 2017.

distribution and obtain $p$ candidate masks from SAM. For each of these masks, we compute the IoU with the original predicted mask $\hat{M}_t^{(x)}$ and select the one with the highest score. Details about IoU computation are in Section A.3. Finally, we extract the logits associated with the selected SAM mask and apply the multi-object prediction procedure described earlier to finalize the segmentation.

## 5 EXPERIMENTS

### 5.1 EVALUATION DATASETS & METRICS

We evaluate our method on four widely used semi-supervised VOS benchmarks: DAVIS-2016 Perazzi et al. (2016), DAVIS-2017 Pont-Tuset et al. (2017), YouTube-VOS 2018 Xu et al. (2018), and Long Videos Liang et al. (2020). Detailed dataset statistics are provided in Section A.2. We report region similarity $\mathcal{J}_{\mathrm{m}}$, contour accuracy $\mathcal{F}_{\mathrm{m}}$, and their average $\mathcal{J}\&\mathcal{F}_{\mathrm{m}}$. The $\mathcal{J}_{\mathrm{m}}$ is the Jaccard index, or intersection-over-union (IoU), which measures the overlap between predicted and ground-truth masks, averaged over all annotated objects and frames. The $\mathcal{F}_{\mathrm{m}}$ is the boundary F-measure, computed as the harmonic mean of boundary precision and recall, also averaged across objects and frames. Finally, $\mathcal{J}\&\mathcal{F}_{\mathrm{m}}$ is defined as the average of $\mathcal{J}_{\mathrm{m}}$ and $\mathcal{F}_{\mathrm{m}}$, providing an overall indicator of segmentation quality. Implementation details are provided in Section A.1.

### 5.2 ANALYSES ON LABEL PROPAGATION THROUGH CROSS-FRAME ATTENTION MAPS

**Self-Attention vs. Cosine Similarity** Figure 3a compares cross-frame affinity maps obtained from (i) the self-attention layers of the diffusion model (ours) and (ii) cosine similarity scores computed directly from raw diffusion features, following (Tang et al., 2023). Using these maps, we propagate the initial GT mask to each subsequent frame, and evaluate the per-frame $\mathcal{J}\&\mathcal{F}_{\mathrm{m}}$ scores on

| Method | DAVIS 2016 | | | DAVIS 2017 | | | YT-VOS 2018 | | | | | Long Videos | | |
|---|---|---|---|---|---|---|---|---|---|---|---|---|---|---|
| | $\mathcal{J}\&\mathcal{F}_\mathrm{m}$ | $\mathcal{J}_\mathrm{m}$ | $\mathcal{F}_\mathrm{m}$ | $\mathcal{J}\&\mathcal{F}_\mathrm{m}$ | $\mathcal{J}_\mathrm{m}$ | $\mathcal{F}_\mathrm{m}$ | $\mathcal{J}\&\mathcal{F}_\mathrm{m}$ | $\mathcal{J}_\mathrm{s}$ | $\mathcal{F}_\mathrm{s}$ | $\mathcal{J}_\mathrm{u}$ | $\mathcal{F}_\mathrm{u}$ | $\mathcal{J}\&\mathcal{F}_\mathrm{m}$ | $\mathcal{J}_\mathrm{m}$ | $\mathcal{F}_\mathrm{m}$ |
| *Zero-shot* | | | | | | | | | | | | | | |
| STC | 74.5 | 74.7 | 74.4 | 67.6 | 64.8 | 70.2 | 65.5 | 66.0 | 67.1 | 59.8 | 69.2 | 26.2 | 26.4 | 26.0 |
| DIFT | - | - | - | 70.0 | 67.4 | 72.5 | - | - | - | - | - | - | - | - |
| DINO | 81.2 | 80.4 | 81.9 | 71.4 | 67.9 | 74.9 | 62.9 | 64.5 | 67.7 | 53.9 | 65.7 | 45.9 | 45.9 | 45.9 |
| **DRIFT(Ours)** | **85.0** | **83.7** | **86.3** | **74.8** | **70.7** | **78.9** | **68.5** | **68.1** | **72.1** | **61.5** | **72.4** | **48.3** | **48.5** | **48.1** |
| *Zero-shot with Image Segmentation Annotations* | | | | | | | | | | | | | | |
| SegGPT | 82.3 | 81.8 | 82.8 | 75.6 | 72.5 | 78.6 | 74.7 | **75.1** | **80.2** | 67.4 | 75.9 | 21.2 | 18.6 | 23.7 |
| SAM-PT$^\dagger$ | 83.1 | 83.2 | 82.9 | 77.6 | 74.8 | 80.4 | 74.0 | 73.3 | 76.0 | 70.0 | 76.7 | 4.1 | 2.1 | 6.0 |
| **DRIFT(Ours)** | **86.6** | **87.2** | **86.2** | **81.3** | **78.8** | **83.7** | **75.3** | 74.5 | 77.3 | **71.0** | **78.3** | **49.1** | **47.7** | **50.4** |
| *Fully Supervised* | | | | | | | | | | | | | | |
| CFBI | 89.4 | 88.3 | 90.5 | 81.9 | 79.1 | 84.6 | 81.4 | 81.1 | 85.8 | 75.3 | 83.4 | 53.5 | 50.9 | 56.1 |
| STCN | 91.6 | 90.8 | 92.5 | 85.4 | 82.2 | 88.6 | 83.0 | 81.9 | 86.5 | 77.9 | 85.7 | 87.3 | 85.4 | 89.2 |
| AOT | 91.1 | 90.1 | 92.1 | 84.9 | 82.3 | 87.5 | 85.5 | 84.5 | 89.5 | 79.6 | 88.2 | 84.3 | 83.2 | 85.4 |
| XMem | 91.5 | 90.4 | 92.7 | 86.2 | 82.9 | 89.5 | 85.7 | 84.6 | 89.3 | 80.2 | 88.7 | 89.8 | 88.0 | 91.6 |
| Cutie-base | - | - | - | 88.8 | 85.4 | 92.3 | 86.1 | 85.5 | 90.0 | 80.6 | 88.3 | - | - | - |

Table 1: **Quantitative Comparisons to SOTA Methods.** Results are reported on Perazzi et al. (2016); Pont-Tuset et al. (2017); Xu et al. (2018); Liang et al. (2020). The subscripts $s$ and $u$ on YT-VOS 2018 indicate seen and unseen categories. *Zero-shot* methods do not use segmentation annotations during training. Methods in *zero-shot with image segmentation annotations* section utilizes models that are trained on large image segmentation datasets. *Fully supervised* models are trained on video segmentation datasets and shown for reference. $^\dagger$SAM-PT is evaluated using CoTracker Karaev et al. (2024), which is pretrained on a video dataset for dense point tracking.

the DAVIS validation set. The results reveal a clear performance gap between the cosine-similarity baseline and our self-attention–based affinity maps. Cosine similarity, which directly compares raw features, is easily influenced by feature components unrelated to the target object. This results in low $\mathcal{J}\&\mathcal{F}_\mathrm{m}$ scores (orange line), caused by dispersed similarity maps, as illustrated in the example propagated masks associated with the orange curve—even for frames near the initial mask where appearance changes are minimal. By contrast, our method exploits the learned projections in the self-attention layers, enabling similarity estimation along semantically meaningful dimensions. The multi-head design further enriches the label-propagation kernel by capturing multiple complementary aspects of similarity. Consequently, cross-frame attention maps yield nearly twice the propagation performance of raw-feature cosine similarity across timesteps (blue line), with examples of accurately propagated masks shown alongside the corresponding blue-curve results.

**Effects of Diffusion Timesteps and DDIM Inversion** We analyze the effect of diffusion timesteps and DDIM inversion on mask propagation performance. Figure 3b illustrates $\mathcal{J}\&\mathcal{F}_\mathrm{m}$ across different timesteps. When injecting white noise, maximum performance is attained at timestep 21 (57.4%) and then quickly degrades as the timestep increases, reflecting the well-known trade-off (Tang et al., 2023; Wang et al., 2024a) that excessive noise at large timesteps washes out original semantics, while at step 1 the nearly noise-free latents show lower $\mathcal{J}\&\mathcal{F}_\mathrm{m}$ than at step 21. In contrast, DDIM inversion—which perturbs latents with model-predicted noise and thus initializes from a model-aligned representation—reaches a higher peak at step 81 (58.0%) and remains above the forward curve at all evaluated timesteps. This consistent advantage supports our premise that DDIM inversion preserves semantic information more faithfully than the standard random noise injection, yielding more reliable features for mask propagation across a broad range of diffusion timesteps.

**Effects of Textual Inversion** In Figure 3c, we compare four prompt types: a null prompt (Null), object class names such as 'dog' or 'person' (Class), BLIP-2–generated object-specific captions (Caption; detailed in Section A.4), and learned embeddings obtained through textual inversion (Gal et al., 2022) (Learned). While Null simply uses an empty prompt as the embedding, it already provides a strong baseline of 71.8% in $\mathcal{J}\&\mathcal{F}_\mathrm{m}$. Supplying text prompts semantically aligned with the target object (Class and Caption) yields only marginal improvements, with absolute gains of 0.1% and 0.2%. In contrast, learning text prompts via textual inversion with our propagation loss achieves a substantial improvement of 3.8%. These results indicate that prompts aligned with object semantics—such as class names or captions—are not the key drivers of performance in this setting, despite

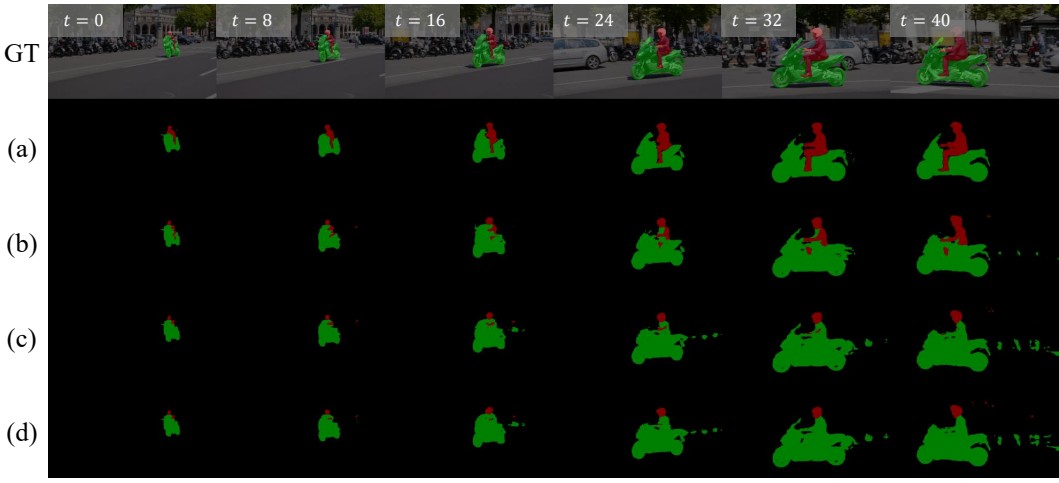

Figure 5: **Qualitative Comparison of Model Variants on DAVIS 2017.** Each row shows segmentation results over time, with the top row showing GT masks and the others corresponding to the model ablations in Figure 6a. Removing components leads to spatial drift or semantic ambiguity, while the full model (a) maintains accurate and coherent instance masks across frames.

|   | Ablations | $\mathcal{J}\&\mathcal{F}_m$ | $\mathcal{J}_m$ | $\mathcal{F}_m$ |
|---|---|---|---|---|
| (a) | DRIFT | 81.3 | 78.8 | 83.7 |
| (b) | $-$SAM | 74.8 | 70.7 | 78.9 |
| (c) | $-$TI&HW | 71.8 | 67.9 | 75.6 |
| (d) | $-$DI | 71.1 | 67.0 | 75.1 |

(a) Effects of each component of DRIFT.

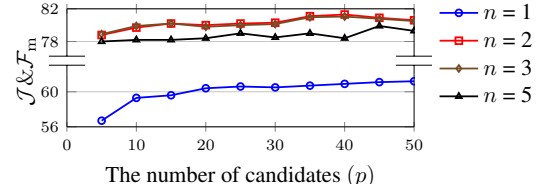

(b) Ablation of SAM prompt configurations.

Figure 6: **Ablation of DRIFT Component and SAM Prompt Configurations** (a) presents ablations by progressively removing DRIFT components, while (b) varies the number of points per prompt ($n$) and the number of mask candidates ($p$) for SAM. All results are reported on DAVIS 2017.

what might be commonly assumed. Instead, embeddings learned specifically for the target task of label propagation yield greater improvements, suggesting that propagation quality can be further enhanced through task-driven or even test-time optimization. However, we observed that textual inversion had a limitation under cosine similarity, and we provide further discussion in Section B.1.

**Distribution of Learned Embeddings** Figure 4a visualizes token embeddings of object class names and textual-inversion–learned embeddings aligned to specific objects. We observe two clearly separated clusters, each grouping one type of embedding. This indicates that the learned embeddings do not encode semantic information about the target object. Instead, they behave as small, learnable parameters that allow fine-grained control over self-attention maps, effectively acting as tunable knobs for label propagation.

**Effect of Adaptive Head Weighting** We also assess the adaptive head weighting technique, comparing it to uniform averaging under two conditions: (i) with DDIM inversion and textual inversion (+DI+TI), and (ii) without them (Baseline). The results show that learned weighting consistently achieves higher $\mathcal{J}\&\mathcal{F}_m$ than uniform averaging. These improvements suggest that combining different heads attending to complementary regions yields more reliable mask propagation.

### 5.3 EVALUATION OF DRIFT ON OBJECT TRACKING VIA SEGMENTATION

**Models & Baselines** We evaluate two variants of DRIFT: a pure diffusion-based version without SAM and a SAM-integrated version. The former is compared against zero-shot baselines STC (Jabri

et al., 2020), DINO (Caron et al., 2021), and DIFT (Tang et al., 2023), while the latter is compared against SegGPT (Wang et al., 2023) and SAM-PT (Rajič et al., 2025), which are zero-shot with image segmentation annotations.

**Comparisons to State-of-the-Art Methods**  Table 1 presents a comparison between the proposed DRIFT and existing state-of-the-art methods. Without SAM (Kirillov et al., 2023), our model outperforms three existing methods that also employ a label propagation approach but utilize different features. While DINO (Caron et al., 2021) demonstrates remarkable performance without direct optimization on segmentation data by leveraging robust feature learning, our method shows even better results, achieving an average relative improvement of 5.94% over DINO. When SAM is incorporated into our framework, the performance is further enhanced, achieving an average relative improvement of 5.59% compared to ours without SAM. Furthermore, our full DRIFT surpasses SegGPT (Wang et al., 2023) and SAM-PT (Rajič et al., 2025), which leverage large-scale image segmentation datasets and SAM, respectively, achieving an average relative improvement of 4.62% and 3.6% across the three short video benchmarks. Both SAM-PT and SegGPT struggle to generalize to longer video sequences showing significant performance drops on Long Videos. In contrast, DRIFT demonstrates its superior ability to maintain temporal coherence and instance identity across extended sequences, without requiring video-specific supervision. Note that SAM-PT leverages additional supervision from a point tracking dataset (Doersch et al., 2022), which offers even denser annotations than video object segmentation labels. Despite this, DRIFT outperforms it across all benchmarks and metrics. Finally, despite being a fully zero-shot approach, our method exhibits comparable scores to some fully-supervised methods, highlighting its remarkable generalization and effectiveness without relying on annotated training data.

**Ablation of Each Component**  The ablation results are consistent with the earlier analysis. Figure 6a quantifies the impact of each component in DRIFT, with qualitative examples provided in Figure 5. Starting from the full model (a), which includes DDIM inversion (DI), textual inversion with adaptive head weighting (TI&HW), and the SAM module (SAM), we observe the highest performance with 81.3% in $\mathcal{J}\&\mathcal{F}_{\mathrm{m}}$, reflecting precise boundary refinement and stable mask propagation. Removing each component leads to performance drops, illustrating the effectiveness of SAM for boundary refinement, textual inversion and head weighting for instance discrimination, and DDIM inversion for semantic stability. The same trend is visible in the qualitative comparisons in Figure 5, where each component contributes to more coherent and temporally consistent segmentation.

**Point Sampling for Prompting SAM**  Finally, we investigate how the contribution of SAM (Kirillov et al., 2023) to the segmentation quality varies with the number of sampled points ($n$) and mask candidates ($p$), as shown in Figure 6b. When $n=1$, performance is poor because a single prompt often produces unstable segments, covering only part of the object or an overly large area. With $n=2$, stability improves substantially, leading to a significant performance gain, which then saturates at $n=3$. Adding more points (*e.g.*, $n=5$) degrades performance, as the increased number of prompts raises the likelihood of including mislabeled regions. Increasing $p$ also improves performance, but the gains saturate at around $p=40$. Overall, with $n=2$ points and $p=40$ candidates, we achieve the best performance of 81.3% in $\mathcal{J}\&\mathcal{F}_{\mathrm{m}}$, with a improvement over 74.8% without refinement.

## 6  CONCLUSION

In this work, we show that diffusion self-attention serves as an effective label propagation kernel for object tracking via segmentation. It provides more robust affinities than cosine similarity, benefits from semantically aligned representations via DDIM inversion, and is further enhanced by test-time strategies such as textual inversion and adaptive head weighting. These results highlight the strong semantic capture ability of diffusion self-attention and its potential as a general tool for video understanding.

## ETHICS STATEMENT

We acknowledge that LLMs were used as writing assistants to improve grammar, clarity, and readability of the manuscript. We also have read and understood the ICLR Code of Ethics and ensured that our work adheres to its principles.

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

# DIFFUSION MODELS ARE TRAINING-FREE OBJECT TRACKERS

## *Supplementary Material*

## A    ADDITIONAL DETAILS

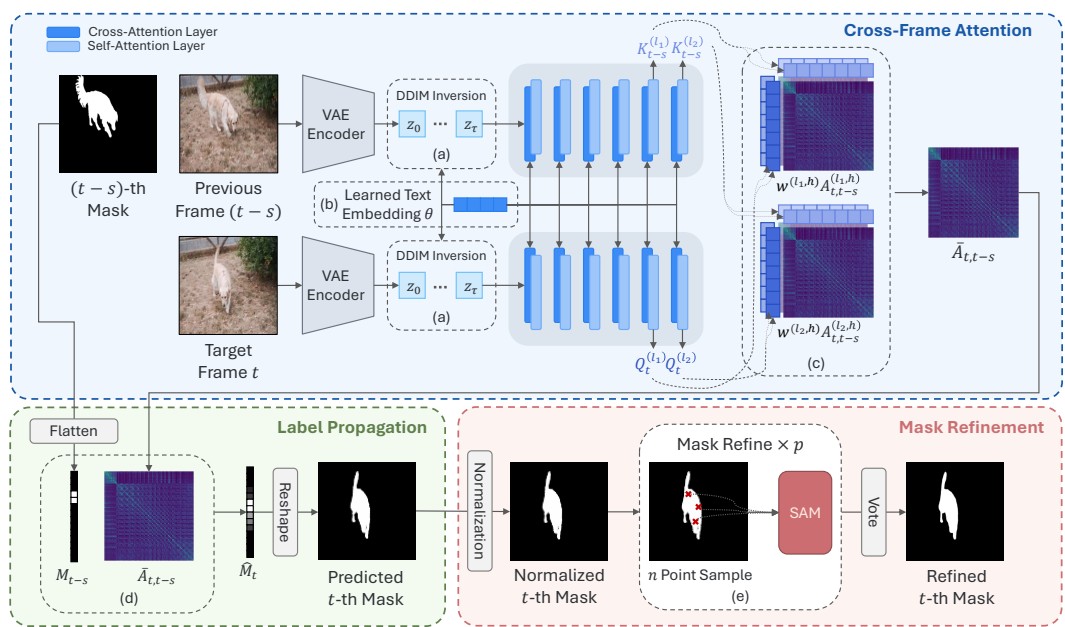

Figure 7: **Overall Pipeline of DRIFT.** Given a target frame $t$ and a reference frame $t-s$ with its mask, (a) we perform DDIM inversion to obtain semantic latent representations $\mathbf{z}_\tau$. (b) Text embeddings $\theta$ are learned for the mask and fed to the diffusion model. Cross-frame attention maps are then computed by matching queries from the target frame with keys from the reference frame across selected layers, and (c) the multi-head attention maps are aggregated using head-specific weights. (d) The aggregated cross-frame attention map $\bar{A}_{t,t-s}$ is used to propagate the mask from frame $t-s$ to $t$. (e) Finally, the obtained soft mask is refined using SAM resulting in a fine-grained mask in the target frame.

### A.1    IMPLEMENTATION DETAILS

We adopt Stable Diffusion 2.1 (Rombach et al., 2022) as our backbone, which takes $768 \times 768$ images and produces $96 \times 96$ latent features. Self-attention maps are extracted from the first attention layer in the final decoder block. At test time, we jointly optimize a mask-specific text token $\theta$ and head weights $w^{(l,h)}$ for each instance using Adam (lr 1e−4, 3,500 steps). DDIM inversion targets timestep $\tau = 41$ using 50 steps from the 1000-step schedule. For refinement, we use SAM (Kirillov et al., 2023) (ViT-H) with $n = 2$ point prompts sampled from the normalized soft mask, generating $p = 40$ candidate masks; the one with highest IoU is selected. Label propagation uses the 7 most recent frames with initial frame. We apply spatial masking with radius $r = 14$ and retain the top $k = 15$ attention scores per query. All experiments are conducted on NVIDIA H100 GPUs.

### A.2    EVALUATION DATASET DETAILS

**DAVIS-2016 Perazzi et al. (2016).**    The DAVIS-2016 dataset was originally introduced for single-object video object segmentation. The validation set consists of 20 high-quality video sequences

(480p resolution), each annotated with a single foreground object mask at every frame. On average, each sequence contains around 70 frames, and the dataset is widely used to benchmark single-object VOS methods due to its precise, per-frame annotations.

**DAVIS-2017 Pont-Tuset et al. (2017).** DAVIS-2017 extends DAVIS-2016 by introducing multiple annotated objects per video, thereby increasing the difficulty of the segmentation task. The validation set contains 30 sequences with a total of 59 annotated objects, with around 70 frames per video. All frames are annotated, and the dataset is widely considered the standard benchmark for multi-object semi-supervised VOS evaluation.

**YouTube-VOS 2018 Xu et al. (2018).** YouTube-VOS is the largest public benchmark for video object segmentation. The 2018 validation set includes 474 video sequences, covering 91 object categories, of which 65 are seen during training and 26 are unseen. Unlike DAVIS, annotations are provided every 5th frame instead of every frame. This yields a total of 12,593 annotated frames in the validation split. The large scale and category diversity of YouTube-VOS make it a challenging and comprehensive benchmark, particularly for evaluating generalization to unseen categories.

**Long Videos Liang et al. (2020).** The Long Videos dataset was designed to test the robustness of segmentation methods on long-duration sequences. It contains 3 validation videos, each with an average length of about 2,470 frames, far exceeding the sequence lengths of DAVIS or YouTube-VOS. For evaluation, 20 frames are uniformly sampled from each video and manually annotated with object masks. This setup allows the benchmark to focus on assessing temporal consistency and robustness of segmentation methods under extended time horizons.

## A.3 SOFT IoU METRIC

To select the most accurate refinement from multiple mask candidates generated by SAM Kirillov et al. (2023), we measure the similarity between each binary SAM mask and the original soft mask prediction using a soft IoU metric. Given a soft mask $A \in [0,1]^{H \times W}$ and a binary candidate mask $B \in \{0,1\}^{H \times W}$, the soft IoU is computed as:

$$\text{IoU}(A, B) = \frac{\sum_{i,j} \min(A_{i,j}, B_{i,j})}{\sum_{i,j} \max(A_{i,j}, B_{i,j}) + \epsilon} \tag{3}$$

where $\epsilon$ is a small constant added to the denominator for numerical stability.

Unlike the standard IoU computation that uses discrete set operations, soft masks represent confidence values or probability distributions over space. Therefore, we interpret the intersection and union between soft and binary masks as element-wise $\min$ and $\max$ operations, respectively. This formulation retains the probabilistic nature of the soft mask while enabling consistent comparison with discrete predictions. Before computing soft IoU, we normalize the soft mask $A$ such that its values sum to one across spatial dimensions, treating it as a spatial probability distribution. To generate $p$ candidate masks, we sample $n$ point prompts from the normalized soft mask $p$ times, each used as a prompt to SAM. The mask with the highest soft IoU score against the original soft prediction is selected as the final output.

## A.4 DETAILS OF BLIP-2 CAPTIONING

We use BLIP-2 (Li et al., 2023) to generate noun-phrase captions for annotated instances in the first frame. Each object is cropped from the DAVIS mask with a small margin, and the background is masked out so that only the target region remains visible. The masked crops are then passed to BLIP-2 with the prompt, `"Question: Provide a short noun phrase that names only the main object in the image.\n Answer:"`. The resulting captions are assigned to the corresponding instances and stored as their initial prompts.

| | Models | Execution Time (s/object) | $\mathcal{J}\&\mathcal{F}_\mathrm{m}$ |
|---|---|---|---|
| 1 | DINO (ViT-B/8) | 11.410 | 71.4 |
| 2 | SAM-PT (CoTracker + SAM-H) | 13.744 | 77.6 |
| 3 | DRIFT (cross-frame propagation only) | 9.853 | 71.1 |
| 4 | + DDIM inversion | + 4.140 | 71.8 |
| 5 | + SAM | + 11.704 | 80.7 |
| 6 | + Textual inversion | + 141.437 | 81.3 |

Table 2: Comparison of execution time and performance across different models and variants.

## B  FURTHER ANALAYSES

### B.1  LIMITATION OF COSINE SIMILARITY UNDER TEXTUAL INVERSION

As illustrated in Figure 2, cosine similarity maps derived from raw diffusion features are noisy, often highlighting irrelevant regions in addition to the target object. Ideally, an affinity measure should emphasize the target region while suppressing unrelated areas, but cosine similarity lacks this selectivity. Moreover, in our setup only the textual tokens are optimized during textual inversion, while the diffusion backbone remains frozen. This limited degree of freedom makes it difficult to correct the noisy propagation induced by cosine similarity, highlighting the advantage of using self-attention maps that already encode semantically structured affinities.

### B.2  COMPONENT-WISE RUNTIME ANALYSIS

We report the average processing time per object on the DAVIS (Perazzi et al., 2016) validation set, measured on a single NVIDIA H100 GPU. Table 2 presents both accuracy and runtime for DRIFT and relevant baselines. Our results are shown in a component-wise manner, where the total runtime corresponds to the sum of the relevant modules. The base propagation step in DRIFT is comparable in speed to existing trackers such as DINO (see rows 1 and 4) and SAM-PT (see rows 2 and 5) , while already achieving similar accuracy. Incorporating DDIM inversion and SAM refinement leads to steady accuracy gains, with runtime overheads of 2.6s and 12.0s per object, respectively. Textual inversion adds a larger overhead, but it is performed only once on the first frame, making its relative cost less dominant for longer sequences. Importantly, even without textual inversion or SAM refinement, DRIFT achieves stronger performance than competing baselines, highlighting the inherent spatio-temporal capability of pretrained diffusion models for objecet tracking via segmentation. While textual inversion increases runtime, the corresponding accuracy improvement demonstrates an acceptable trade-off in practice.

### B.3  COMPARISON ACROSS DIFFUSION MODEL VARIANTS

| Model | Params | SAM | $\mathcal{J}\&\mathcal{F}_\mathrm{m}$ | $\mathcal{J}_\mathrm{m}$ | $\mathcal{F}_\mathrm{m}$ |
|---|---|---|---|---|---|
| SD 1.5 | 860M | ✗ | 68.9 | 65.6 | 72.1 |
| | | ✓ | 76.4 | 74.1 | 78.8 |
| SD 2.1 | 865M | ✗ | 74.8 | 70.7 | 78.9 |
| | | ✓ | 81.3 | 78.8 | 83.7 |

Table 3: **Comparison of SD 1.5 and 2.1 with and without SAM on DAVIS 2017.** This ablation highlights the effect of backbone diffusion model and shows how performance varies with and without SAM-based refinement.

Besides our primary results based on Stable Diffusion 2.1, we also evaluate our framework using Stable Diffusion 1.5 (Rombach et al., 2022) as the backbone. This variant processes $512 \times 512$ resolution inputs and produces latent features of size $64 \times 64$. We extract self-attention maps from all three attention layers in the final decoder block of the U-Net. All other settings remain identical

to those used with the 2.1 backbone. As shown in Table 3, we observe that performance trends remain consistent, confirming the general applicability of our method across diffusion model versions. Notably, the performance disparity between SD 2.1 and SD 1.5 backbones is not attributed to model size, as their U-Net parameter counts are nearly identical (865M vs. 860M), suggesting that other factors, such as differences in training data or procedures, play a more significant role.

### B.4 MASK REFINEMENT WITH PAC-CRF

| Model | Refiner | $\mathcal{J}\&\mathcal{F}_{\mathrm{m}}$ | $\mathcal{J}_{\mathrm{m}}$ | $\mathcal{F}_{\mathrm{m}}$ |
|---|---|---|---|---|
| Uziel et al. (2023) | ✗ | 74.1 | - | - |
| | PAC-CRF | 76.3 | **73.8** | 78.7 |
| DRIFT(Ours) | ✗ | 74.5 | 70.3 | 78.6 |
| | PAC-CRF | **76.4** | 73.0 | **79.8** |
| | SAM | 81.3 | 78.8 | 83.7 |

Table 4: **Comparison of Refinement Method on DAVIS 2017.** All models are based on SD 2.1 with DDIM inversion and textual inversion applied. Except for the refinement module, all other experimental settings are kept identical.

In addition to our primary refinement method using SAM, we also explore the use of PAC-CRF (Su et al., 2019) as a lightweight post-processing technique for enhancing the spatial quality of predicted masks. PAC-CRF refines a segmentation mask by enforcing local smoothness and edge-aware consistency using the underlying image as guidance. Following prior work (Su et al., 2019; Uziel et al., 2023), we apply PAC-CRF with a kernel size of $5 \times 5$ and 30 refinement steps to binary masks $\hat{M} \in \{0, 1\}^{H \times W}$, treating them as noisy initial labels, and use the corresponding image $I \in \mathbb{R}^{H \times W \times 3}$ to guide the refinement via pairwise potentials that penalize label inconsistencies between neighboring pixels with similar appearance. While not as powerful as prompt-based refinement with SAM, PAC-CRF can moderately improve mask alignment near object boundaries with low computational overhead. Quantitative results comparing refinement strategies are presented in Table 4.

## C LIMITATIONS

A limitation of our approach is the computational overhead of textual inversion. This step, though required only once per object on the first frame, involves optimizing prompt embeddings and can be costly when applied to datasets with many videos or object instances. All experiments were conducted on NVIDIA H100 80GB GPUs with the diffusion model kept frozen. While textual inversion entails additional computation, memory usage remains modest throughout both the inversion and inference stages. Reducing the overhead of textual inversion—through faster optimization, caching strategies, or amortized prompt learning—remains an important direction for future work.

