# OpenReview forum: "Diffusion Models Are Training-Free Object Trackers"
_ICLR.cc/2026/Conference — ICLR 2026 Conference Withdrawn Submission_

### Official Review · Reviewer_FAvV · 2025-10-17

**Soundness:** 2
**Presentation:** 3
**Contribution:** 1
**Rating:** 2
**Confidence:** 3

**Summary:**

This paper proposes a new training-free video object tracking method called DRIFT. The authors discover that the self-attention layers of diffusion models inherently possess a semantic label propagation capability, allowing pixel-level semantic consistency to be transferred across frames, thus enabling zero-shot object tracking and segmentation. The method constructs cross-frame self-attention to establish a temporally consistent label propagation kernel and integrates three test-time optimization strategies—DDIM inversion (for semantically aligned latent representations), textual inversion (to learn object-specific text embeddings), and adaptive head weighting (for adaptive multi-head attention aggregation)—which significantly improve tracking accuracy. Experiments on standard benchmarks such as DAVIS and YouTube-VOS demonstrate strong performance, validating the powerful potential of diffusion model self-attention for semantic understanding and video tracking.

**Strengths:**

**Clear writing and presentation.** The paper is clearly written and logically structured, making it easy to follow. Figures and tables are well-designed and effectively support the main arguments. Overall, the presentation quality enhances the readability and credibility of the work.

**Comprehensive ablation studies.** The paper provides extensive ablation experiments that thoroughly examine each component of the proposed framework. These studies convincingly demonstrate the contribution and necessity of key modules such as DDIM inversion, textual inversion. The completeness of the analysis strengthens the empirical validity of the paper’s claims.

**Weaknesses:**

**Limited methodological novelty.** The core idea of *cross-frame self-attention map propagation* presents limited methodological novelty. This mechanism is already a fundamental component in video diffusion models through *full-frame self-attention*. It would be valuable for the authors to discuss whether employing a video diffusion model directly could achieve similar or even better results.

**Outdated backbone model.** The paper employs **Stable Diffusion 2.1** (proposed in 2022) as the backbone, which is now considered relatively outdated for image generation tasks. Using such a model makes it difficult to verify whether the proposed approach remains effective under more recent diffusion architectures or paradigms. Adopting newer models could better demonstrate the robustness and general applicability of the method.

**Inaccurate claim of “training-free” and “zero-shot”.** The *textual inversion* process involves optimizing learnable embeddings for each individual case, which can be regarded as a form of per-instance training. Therefore, the claim that the method is entirely “training-free” and “zero-shot” may require further clarification or qualification.

**Insufficient comparison.** A considerable portion of the performance gain appears to stem from the **SAM-based mask refinement** module. Comparisons with additional SAM-based refinement methods would help better isolate the specific contribution of the proposed diffusion-based tracking framework. Including such results could make the empirical analysis more comprehensive and convincing.

**Questions:**

Refer to the four points in weakness.

---

### Official Review · Reviewer_TLKP · 2025-10-28

**Soundness:** 3
**Presentation:** 3
**Contribution:** 3
**Rating:** 6
**Confidence:** 4

**Summary:**

The paper shows that self-attention in pretrained text-to-image diffusion models can be reinterpreted as a label-propagation kernel for training-free video object tracking via segmentation. It builds a cross-frame attention operator to propagate first-frame masks and augments it with three test-time procedures: DDIM inversion (to obtain semantically aligned latents), textual inversion of a learnable prompt token tied to the object mask, and adaptive head weighting across attention heads. The full system, DRIFT, optionally refines propagated masks with SAM and claims SOTA among zero-shot methods on DAVIS-16/17, YouTube-VOS’18 and Long Videos, sometimes approaching supervised trackers

**Strengths:**

1. The paper is well-written and clearly structured, making it easy to follow the methodology and understand the contributions.


2. The ablations and analysis are thorough, isolating self-attention vs cosine similarity, the effect of DDIM inversion across timesteps, prompt types (null/class/caption/learned), and learned head weights; includes per-frame curves and visualizations.


3. The zero-shot results are competitive, improving over strong propagation baselines (STC, DINO, DIFT), and further gains when adding SAM.

**Weaknesses:**

1. The "training-free" claim is somewhat overstated, as the method involves several optimization steps at test time (textual inversion), which can introduce significant computational overhead and latency, limiting the feasibility for real-time applications. Moreover, the specific training details and processes are not adequately described, requiring more information to assess the actual training complexity and resource requirements.


2. Table 1 needs to compare with more recent and stronger training-free baselines, such as Matcher[1].

[1] Matcher: Segment Anything with One Shot Using All-Purpose Feature Matching. ICLR2024

3. The use of self-attention in Diffusion Models has already been applied in editing or segmentation tasks, such as "Towards Understanding Cross and Self-Attention in Stable Diffusion for Text-Guided Image Editing" and "Unleashing the Potential of the Diffusion Model in Few-shot Semantic Segmentation". The connections and distinctions between these works and the current paper need to be further clarified.
[2] Towards Understanding Cross and Self-Attention in Stable Diffusion for Text-Guided Image Editing. CVPR2024
[3] Unleashing the Potential of the Diffusion Model in Few-shot Semantic Segmentation. NeurIPS2024

**Questions:**

1. I am curious whether the self-attention in SD3, which is based on the DIT architecture, has any advantages.
2. Why not use a text-to-video diffusion model for video object tracking? Would this lead to better temporal consistency?
3. What are the advantages of the training-free method Drift compared to SAM2?
4. The abstract on OpenReview needs to be updated. ( \drift{})

---

### Official Review · Reviewer_cLfF · 2025-10-30

**Soundness:** 2
**Presentation:** 3
**Contribution:** 2
**Rating:** 4
**Confidence:** 2

**Summary:**

This paper proposes a traning-free object tracking method based on diffusion models. This work leverages self-attention maps instead of features to provide semantic labels, enabling zero-shot object tracking via segmentation without training. To further enhance the attention maps across frames, test-time optimizations are introduced, including DDIM inversion, textual inversion, and adaptive head weighting. The authors propose DRIFT, combining test-time optimizations and SAM, to achieves state-of-the-art performance in zero-shot object tracking.

**Strengths:**

- The paper has a good writing flow to present the motivation and problem settings.
- The discussion about feature similarity and attention map for semantic labels is insightful, which could lead to deeper understanding of ViTs for capturing objects.
- The proposed test-time optimizations clearly demonstrate their effects by ablation study and visualization.

**Weaknesses:**

- It is tricky for a work is not free of training but training-free. This paper did not show the number of parameters, such as prompt embedding and head weights, and the cost for training. Therefore, the comparison to SMITE is desired to demonstrate the advantages or disadvantages in efficiency.
- The baseline models are not convincing. For instance, dino is an image pretrained model on IN1K, which went through fewer images than SD. DINOv2 or v3 should be considered for fair comparison as feature similarity vs. attention map. Second, the enhancement requires much more runtime to segment. DINO could gain better performance by Feature Upsampling [1] or using a large model in under the restricted runtime.
- lacking discussion of Attention-based Semantic Segmentation, such as  [2].
- the major improvement is the combination of SAM. It is hard to prove that diffusion models have learned better object information.

[1]: Wimmer, Thomas, et al. "AnyUp: Universal Feature Upsampling." arXiv preprint arXiv:2510.12764 (2025).
[2]: Huang, Jianqiang, et al. "Attention-based class activation diffusion for weakly-supervised semantic segmentation." arXiv preprint arXiv:2211.10931 (2022).

**Questions:**

What is the training cost and comparison to SMITE?

---

### Official Review · Reviewer_QiHn · 2025-10-31

**Soundness:** 2
**Presentation:** 2
**Contribution:** 2
**Rating:** 4
**Confidence:** 3

**Summary:**

The paper tackles the problem of object tracking. The authors introduce a zero-shot framework, DRIFT, that leverages self-attention maps from pretrained text-to-image diffusion models to perform the task. For refinement, DRIFT integrates SAM to improve mask boundaries.

**Strengths:**

1. The paper proposes an interesting perspective: extending self-attention module across time for object tracking.
2. The finding that self-attention maps of diffusion models encode dense correspondences is compelling and well-illustrated.

**Weaknesses:**

1. Leveraging pre-trained diffusion model and its attention maps for object tracking and segmentation is an extensively studied problem. Although the authors claim that their focus is on self-attention module of diffusion model, its advantages over existing methods are not clearified, making the main technical contribution somewhat incremental to exsiting ones.
2. The method highlights "training-free", but involves test-time optimization, which appear to have large contribution to the performance (Fig. 6a). The term training-free is thus misleading. Optimization with thousands of steps (3,500, as shown in the appendix) is computationally heavy and data-specific. The paper should clarify this distinction between "no pretraining" and "per-instance optimization".
3. Due to the test-time optimization, the time cost for DRIFT to perform each example is quite large, taking >2 min as shown in Tab. 2. Also, if given comparable runtime with SAM-PT (14 sec), DRIFT (e.g., DRIFT (cross-frame propagation only) + DDIM inversion) performs much worse than SAM-PT.
4. The most visible accuracy gains arise primarily from SAM mask refinement, which has little to do with the main claimed contribution of this paper. Without SAM, DRIFT performs comparably to earlier attention-propagation methods, weakening the claim that diffusion alone yields state-of-the-art zero-shot tracking.

**Questions:**

Please refer to the weakness. My major concern is with the technical contribution of this paper.

---

### Official Review · Reviewer_8TiL · 2025-11-06

**Soundness:** 3
**Presentation:** 3
**Contribution:** 2
**Rating:** 4
**Confidence:** 3

**Summary:**

This paper explores leveraging diffusion models for training-free zero-shot object tracking via segmentation. By reinterpreting the self-attention maps of pretrained text-to-image diffusion models as semantic label propagation kernels and extending them across frames, the authors achieve temporal label propagation without task-specific training. Additionally, the authors find that self-attention maps outperform raw diffusion feature cosine similarity for capturing robust semantic affinities, while test-time optimizations (DDIM inversion, textual inversion, adaptive head weighting) and SAM integration further enhance mask accuracy. The proposed DRIFT framework is thoroughly described, with detailed ablation studies validating the contribution of each component. Experiments on standard VOS benchmarks demonstrate that DRIFT outperforms previous zero-shot methods and even achieves competitive results with some supervised approaches, showcasing strong effectiveness and generalization.

**Strengths:**

1. The paper provides intuitive analyses of the characteristics of diffusion models' feature maps and conducts appropriate ablation studies for the proposed improvements.
2. The description of the proposed method is highly detailed and comprehensive.
3. The writing is clear and easy to understand.

**Weaknesses:**

1. Table 2 in the Supplementary Material raises certain concerns. From the table, it can be observed that SAM contributes the most significant performance gain. The proposed DRIFT framework mainly leverages diffusion models for similarity calculation and coarse mask propagation. What advantages does it offer compared to methods that directly perform mask propagation (e.g., CoTracker)? Why does the performance with SAM in Table 2 outperform SAM-PT by such a large margin? Does the main improvement stem from the proposed method itself or the adaptive optimization of SAM?
2. In Table 1, the proposed method approaches the performance of some classic fully supervised methods (e.g., STCN) on the first three benchmarks but lags far behind on Long Videos—a scenario with greater practical application value. This issue requires additional analysis.

**Questions:**

1. Regarding textual inversion, is my understanding correct that it is necessary to optimize a text token embedding for segmentation based on the input frame and GT mask of the first frame in each case?
2. In Table 2 of the Supplementary Material, adding textual inversion incurs substantial computational time but yields relatively minimal performance improvement. In this ablation study, if textual inversion is not used, how are the text token embeddings initialized? Compared to Figure 3(c) in the main text, why is there such a significant discrepancy in the observed gains?

---

### Note · Authors · 2025-11-13

I have read and agree with the venue's withdrawal policy on behalf of myself and my co-authors.